# Interface Design in Lightweight SiC/TiSi_2_ Composites Fabricated by Reactive Infiltration Process: Interaction Phenomena between Liquid Si-Rich Si-Ti Alloys and Glassy Carbon

**DOI:** 10.3390/ma14133746

**Published:** 2021-07-04

**Authors:** Donatella Giuranno, Sofia Gambaro, Grzegorz Bruzda, Rafal Nowak, Wojciech Polkowski, Natalia Sobczak, Simona Delsante, Rada Novakovic

**Affiliations:** 1Institute of Condensed Matter Chemistry and Technologies for Energy (ICMATE), National Research Council of Italy (CNR), Via De Marini 6, 16149 Genoa, Italy; sofia.gambaro@ge.icmate.cnr.it (S.G.); simona.delsante@unige.it (S.D.); rada.novakovic@ge.icmate.cnr.it (R.N.); 2Lukasiewicz Research Network-Krakow Institute of Technology, Zakopiańska 73 Str., 30-418 Krakow, Poland; grzegorz.bruzda@kit.lukasiewicz.gov.pl (G.B.); rafal.nowak@kit.lukasiewicz.gov.pl (R.N.); wojciech.polkowski@kit.lukasiewicz.gov.pl (W.P.); n.sobczak@imim.pl (N.S.); 3Institute of Metallurgy and Materials Science, Polish Academy of Sciences (IMMS-PAS), 25 Reymonta Str., 30-059 Krakow, Poland; 4Department of Chemistry and Industrial Chemistry, Genoa University and Genoa Research Unit of INSTM, Via Dodecaneso 31, 16146 Genoa, Italy

**Keywords:** MMCs, CMCs, aerospace, wetting at high temperature, Ti-silicides

## Abstract

To properly design and optimize liquid-assisted processes, such as reactive infiltration for fabricating lightweight and corrosion resistant SiC/TiSi_2_ composites, the extensive knowledge about the interfacial phenomena taking place when liquid Si-rich Si-Ti alloys are in contact with glassy carbon (GC) is of primary importance. To this end, the wettability of GC by two different Si-rich Si-Ti alloys was investigated for the first time by both the sessile and pendant drop methods at *T* = 1450 °C. The results obtained, in terms of contact angle values, spreading kinetics, reactivity, and developed interface microstructures, were compared with experimental observations previously obtained for the liquid Si-rich Si-Ti eutectics processed under the same operating conditions. As the main outcome, a different Si content did not seem to affect the final contact angle values. Contrarily, the final developed microstructure at the interface and the spreading kinetics were observed as weakly dependent on the composition. From a practical point of view, Si-Ti alloy compositions with a Si content falling in the simple eutectic region of the Si-Ti phase diagram might be potentially used as infiltrating materials of C- and SiC-based composites.

## 1. Introduction

Industrial sectors belonging to lightweight transportation and aerospace continue to base their research and innovation plans on three fundamental pillars: lighter weight, increased strength, and enhanced heat and corrosion resistances [1]. The motivations behind these pillars lie in improving fuel economy through efficiency and lightweighting, in reducing air and spacecraft manufacture costs, and in making air and space travel more cost-effective and safer modes of transportation [2].

In this context, the materials currently gaining the highest interest are metal- and ceramic-matrix composites (MMCs and CMCs) reinforced by high strength continuous ceramic fibers, freshly machinable metals, etc. Moreover, materials of new concepts are emerging and drawing attention as ideal replacements for the less “virtuous” and more traditional structural materials (i.e., Ni-based superalloys, steels, etc.), such as bulk metallic glasses and high entropy alloys [3,4,5,6].

Today, for aerospace and lightweight transportation, the most extensively studied materials are CMCs reinforced by C- and SiC fibers (C_f_ and SiC_f_, respectively), namely C_f_/C, C_f_/SiC, SiC_f_/SiC, and C_f_/C-SiC composites as well as SiC_p_ (SiC particles as reinforcement) [1,3,4,7,8,9,10,11,12]. 

Despite the fact that the manufacturing processes of fibers and reinforcements have reached a high level of reproducibility, their use is limited by the fabrication costs and technological problems encountered in successfully producing large and complex CMC shapes. In this regard, the key steps to take are the assembly of CMCs and their integration/joining with dissimilar materials, i.e., metals, ceramics, or other composites, as well as the prevention of fibers from the degradation occurring during fabrication processes and in service, mainly observed at high temperatures. Indeed, C_f_ and SiC_f_ show the tendency to be oxidized and degraded (i.e., by releasing CO_(g)_ and SiO_(g)_) if processed at temperatures above 400 °C and 1300 °C under oxidizing atmospheres, respectively [13,14]. 

Several papers available in the literature introduce metal silicides as promising densifiers or filler phases for CMCs for improving thermal barriers [8,15] and as structural materials and components for assembling re-entry space vehicles [10,16] and fission/fusion nuclear reactors [17,18,19]. This is due to their peculiarities, such as chemical inertness; improved electrical and thermal stability; self-protection and self-healing capabilities due to their excellent oxidation resistance, mainly at high temperatures; and, for some of the applications mentioned above, their low weight, as in the case of Ti-silicides.

To succeed in the fabrication/joining of such composites, the spontaneous/pressureless infiltration of liquid Ti- or Si-based alloys can be considered as a costless alternative with respect to conventional sintering processes. In addition, if compared with sintering techniques, Si-based alloy infiltration into C- and SiC-based porous materials shows significant benefits. It is a classical liquid-assisted process based on the reaction bonded/formed silicon carbide mechanisms (e.g., RBSC/RFSC), well known to evolve at lower processing parameters (i.e., temperature, pressure, and time) with nearly net shape fabrication capabilities [20,21]. On the other hand, for liquid-assisted processes applied for densifying/joining CMC matrixes, the pivotal issue is to preserve the starting fibers’ microstructure and, consequently, their thermomechanical response. In other words, it lies in the successfully control of the interaction phenomena taking place at the metal/fiber interfaces. Key requirements for obtaining long-term stability between the liquid metal phase and fibers are chemical-physical compatibility, in terms of good wetting/adhesion but limited reaction phenomena, a comparable thermal expansion coefficient (CTE), etc. In addition, to avoid fiber degradation, a suitable set of process parameters needs to be defined, such as a reduced oxygen content inside the surrounding atmosphere and working temperatures ≤ 1450 °C. 

A few previous attempts to use Si-rich Si-Ti alloys (hereafter Si-Ti alloys), including liquid TiSi_2_ and Si-rich eutectic (i.e., Si-16at%Ti) alloys, as brazers and densifying materials for CMCs are available in the literature [18,19,20,21,22,23,24,25,26,27,28,29], even reported by the authors of [30]. Specifically, the use of liquid Si-Ti eutectics for joining monolithic SiC and C_f_/C, C_f_/C-SiC, and SiC_f_/SiC composites at T ≤ 1420 °C without any or negligible degradation of the fiber/matrix interfaces in the composite materials is reported in [23,26,30]. Moreover, the joints produced showed an absence of defects or debonding phenomenon, and a well-adherent layer appeared at the metal matrix/fiber interphase. Specifically, as reported in [26,30], a Si–TiSi_2_ eutectic microstructure constituting the brazing seam, as driven by the cooling process, combined with the appearance of a dense SiC layer well adherent to C and SiC fibers and an overall improvement of the mechanical properties of the joints were documented. 

C_f_/C composites were brazed successfully at T ≥ 1490 °C using directly liquid TiSi_2_ as filler alloy by Dadras et al. [31], and a maximum shear strength of 34.4 MPa at T = 1164 °C was measured at the joint, confirming the fair capabilities of using intermetallic alloys as CMC fillers/brazers for applications at high temperatures. In addition, as the authors highlighted, at the bond interlayer of samples joined by TiSi_2_ interlayers, TiC and SiC were revealed together with the unreacted TiSi_2_, most probably due to the uncompleted reaction between TiSi_2_ and C. 

Aiming to optimize the melt infiltration process, the higher melting point [32] and less fluidity [33] of Ti-silicide with respect to Si-Ti rich alloys makes these intermetallic alloys less attractive for liquid-assisted processes applied for densifying and joining techniques based on Si-Ti alloys. In addition, intermetallic phases, and in particular silicides, do not “emerge” for their fracture toughness values, which are in any case higher than pure Si [34]. On the contrary, silicides show improved thermomechanical properties and oxidation resistance at high temperatures [35]. Moreover, the decrease of process temperatures (*T* ≤ 1420 °C) and a better control of the microstructure evolution are easily obtained by moving the selection of infiltrating alloy to the Si-rich side of the Si-Ti phase diagram. For these reasons, liquid Si-Ti alloys seem to be a good compromise as infiltrating materials for C and SiC porous materials. However, in view of favoring the Si-replacement, the selection of the Si-Ti alloy must be carefully evaluated. Furthermore, aiming to scale up the infiltration process to the industrial level, several factors affecting the efficiency of the reactive infiltration process should be carefully taken into account. Indeed, the starting metal materials may be nonhomogeneous, and a Si enrichment at the surface might be most probably favored, as well as a TiSi_2_ precipitation into the alloy bulk, during alloy manufacturing under not well-controlled atmospheres. Consequently, due to the presence of a large amount of pure Si and silicide precipitates, anomalous melting can be observed. Moreover, the temperature selected during infiltration is usually around 1450 °C, with a heating rate even slower than 5 °C/min, and the selected lower-melting Si-based alloys may melt and infiltrate before reaching the imposed working temperature. 

With such motivations, the influence of Si content on the interaction phenomena taking place when a liquid Si-rich Si-Ti alloy is in contact with GC at *T* = 1450 °C was investigated by the sessile drop method, and the main relevant results are reported in the present work. 

To check the reliability of the results obtained, the interfacial phenomena observed on GC in contact with two liquid Si-rich Si-Ti alloys, in terms of wetting kinetics and developed interface microstructures, were compared with the results previously obtained in the liquid Si-16.4 at%Ti/GC [36] (i.e., Si-rich Si-Ti eutectics) and Si-16.4 at%Ti/SiC [37] systems. Specifically, the Si-Ti hypoeutectic (Si-24 at%Ti) and hypereutectic (Si-8at%Ti) alloys were selected and put in contact with GC at *T* = 1450 °C, and the wetting kinetics were analyzed in terms of contact angle values as a function of time, as well as the formed reaction products and developed interface microstructures. 

Light microscopy (LM) and scanning electron microscopy (SEM), combined with energy dispersive X-ray spectroscopy (EDS) were used to perform the interface microstructure characterization. 

For the sake of clarity, the scope of the paper is to bring out the advantages of preliminary studies on wettability and interaction phenomena in optimizing the melt infiltration process, mainly by addressing the selection of the most suitable set of operating parameters. In other words, by focusing on interfacial phenomena in terms of adhesion, reactivity, growth of reaction layers, etc., some factors negatively affecting the reactive infiltration process, (i.e., pore narrowing/pore closure and/or fiber degradation by C dissolution) can be easily predicted and limited to a large extent.

## 2. Wetting Tests: Experimental Details

### 2.1. Materials and Sample Preparation

GC plates (12 × 12 × 3 mm^3^,) provided by Alfa Aesar© (Karlsruhe, Germany) were selected and used for wettability studies. At the GC substrate surface, a value of roughness Ra ≈ 20 nm was measured by an optical confocal-interferometric profilometer (Sensofar S-neox, Terrassa, Spain)).

The Si-Ti alloys were prepared by mixing and arc melting nominal weights of high purity Si and Ti (99.98%-Goodfellow^®^). To ensure the homogeneity of their composition, the alloy samples were arc melted more than 3 times under an Ar atmosphere (N60, O_2_ < 0.1 ppm). In addition, to further decrease the residual oxygen content inside the arc melting chamber, a Zr-drop was previously melted. By checking the alloys’ final weight, as well as their actual composition by EDS analysis, no evidence of loss of material by evaporation was observed. 

The Si-8at%Ti and Si-24at%Ti alloys of 0.06–0.1 g were successfully obtained, and the as-produced Si-Ti alloys’ microstructure and composition were inspected at the cross-sectioned samples by SEM/EDS, as shown in Figure 1.

A strong Si-segregation was detected at the top of the Si-8at%Ti solidified drop (Figure 1a). At the Si-8at%Ti cross-sectioned sample, pure Si in the form of elongated needle-shaped crystals embedded into a Si-Ti eutectic phase was revealed (Figure 1c). Contrarily, globular TiSi_2_ crystals were observed at the top of the Si-24 at% Ti solidified drop and even inside the alloy bulk as precipitates englobed in a Si rich Si-Ti eutectic microstructure. If compared with the Si-16.2 at% Ti alloy previously studied [30,36,37], identified as the Si-rich Si-Ti eutectic alloy [30], such observations are in full agreement with hypo- and hyper- eutectics, as it is the case for Si-24at%Ti and Si-8at%Ti alloys, respectively [32]. 

Before the wetting experiments, both the alloy sample and the substrate were weighted, rinsed in an ultrasonic bath of isopropyl alcohol, and dried with compressed air.

### 2.2. Sessile Drop Experiments: Devices and Procedures

The Si-Ti alloy/GC couple, assembled at room temperature, was inserted into the experimental device and moved to the central part of the heater.

Prior the experiments, the device was degassed under vacuum (P_tot_ ≤ 10^−6^ mbar) for two hours. Wetting tests were carried out by the classical sessile drop method (SD), accompanied with contact heating of metal/substrate couple, under a static Ar atmosphere (99.9999%, O_2_ < 0.1 ppm) and by following the procedure detailed elsewhere [30]. It worth highlighting that the presence of graphite as heating element provides an atmosphere with reduced oxygen content, as detailed in [30]. 

Wetting experiments were performed by fast heating the sample up to the selected testing temperature (kept constant for 15 min). To preserve the developed interface microstructure as much as possible, the sample was “quenched” (cooling rate about 20 °C/s) at the end of the dwell time by turning off the power to the heater. 

The evolution of wetting kinetics, in terms of contact angle and drop geometric variables (R-base radius and H-drop height) values was in real time followed and recorded by an image analysis software ad hoc, developed in LABVIEW environment (ASTRAVIEW^®^ [38]), and connected to a high-speed CCD camera

To check the reliability of the results obtained, the experimental device described in detail in [39], was used in parallel to process and study the interaction phenomena occurring at the Si-24at%Ti/GC interface. Namely, wettability studies by improved sessile drop method accompanied with noncontact heating of a metal/substrate couple and capillary purification of a metal drop (hereafter dispensed drop (DD)) were done at *T* = 1450 °C under a static Ar atmosphere [40]. 

At the testing temperature of *T* = 1450 °C, the oxygen partial pressure value of PO_2_ < 10^−14^ mbar was achieved by the presence of Ta as heating element and acting as an oxygen getter [41]. Such methodology allows excluding possible factors affecting the kinetics, mainly during the early stage of wetting, such as the presence of native oxide segregated at the alloy surface and delay in the melting due to a nonhomogeneous starting alloy material. On the other hand, the absence of native oxide puts the molten surface directly in contact with the surrounding experimental environment, as well as with the tip of the capillary. Consequently, the molten phase is more prompt to evaporate or to be polluted by gas/liquid interactions and by reaction with the container material. 

The GC substrate and the Al_2_O_3_ capillary filled by pieces of Si-24at%Ti alloy, were loaded into the device at room temperature. The experimental device was heated up to 800 °C under vacuum with a temperature gradient of 5 °C/min. The second heating stage was performed under a static Ar atmosphere and the temperature was increased up to 1500 °C for the complete melting of TiSi_2_ precipitates [32]. Finally, by decreasing the temperature down to 1450 °C, the testing conditions were achieved, and the molten alloy was squeezed and placed onto the GC substrate. 

At the end of the isothermal time (*t* = 15 min), the device was cooled from *T* = 1450 °C down to room temperature with a rate of 20 °C/min. 

Every single frame was processed by ASTRAVIEW^®^ software and both contact angle values and drop geometric parameters simultaneously measured to evaluate wettability and spreading behaviors over time. 

By analyzing the experimental methods/procedures applied and all the factors potentially affecting the reliability of the results, an accuracy of the contact angle data around ±2° is estimated [42].

### 2.3. Surface and Microstructural Characterization

After the wetting tests, all the Si-Ti/GC samples were embedded in cold epoxy-resin, cross sectioned, metallographically polished with SiC papers and diamond pastes, and coated with Au for microstructural characterization. 

As already introduced, in order to analyze the microstructure and reaction products, both at the top and at the cross-sectioned solidified drops, a light microscope (ZEISS, model) and a scanning electron microscope (SEM, Leo 1450 VP, INCA Energy 300) equipped with energy dispersive X-ray spectroscopy (EDS, AX10) were used. 

## 3. Results

### 3.1. Wettability of GC by Si-Rich Si-Ti Alloys as a Function of Si-Content

In Figure 2, the evolution of contact angle values for the Si-8at%Ti/GC and Si-24at%Ti/GC systems are shown and compared with the wettability of GC by the liquid Si-16.2at%Ti eutectic alloy processed under the same operating conditions (i.e., *T* = 1450 °C) under a static Ar atmosphere [30]). 

As it can be seen, the achievement of the steady-state conditions seems to be influenced by the different Si content (Figure 2a,b). Indeed, during the early stages of alloy spreading, the triple line of each Si-Ti/GC system exhibits a different kinetics behavior. In fact, the no-wetting to wetting transition was observed at 6, 10, and 18 s for Si-8at%Ti/GC, Si-16.2at%Ti/GC, and Si-24at%Ti/GC couples, respectively, as evinced in Figure 2b and by the time sequence images shown in Figure 3. Moreover, after the first spreading stage, a decrease in the slope of the wetting kinetics for all the couples was observed. Specifically, the decrease in the rate of spreading comes to be evident after 23, 27, and 30 s after the detected Si-8at%Ti, Si-16.2at%Ti, and Si-24at%Ti melting, where the related contact angle values of θ ≈ 46°, 50°, and 52° were measured. Subsequently, a further decrease of ≈2° in the contact angle value was observed for 7 s at the Si-8at%Ti/GC triple line. Contrarily, the achievement of the steady-state conditions took a longer time in the Si-24at%Ti/GC system, namely 60 s to exhibit a contact angle value of ≈46°, which was kept constant until the end of the dwell time. 

After a time of contact equal to *t* = 900 s between the liquid phase (Si-Ti alloys) and the solid substrate (GC) at *T* = 1450 °C, the contact angle values measured were θ ≈ 41°, 44°, and 46° for the Si-8at%Ti/GC, Si-16.2at%Ti/GC, and Si-24at%Ti/GC couples, respectively. 

Figure 2c shows the triple lines of the three Si-rich Si-Ti alloys/GC samples after the wetting experiments. The “halo” surrounding the drop perimeters, markedly visible for the Si-richer systems, is attributed to the presence of a thin SiC layer, as detected by SEM-EDS analysis, similar to other Si-rich-based alloys processed under the same operating conditions [36,37,43,44,45,46,47], as shown in Figure 4. 

Specifically, a continuous SiC layer accumulated on the GC surface was detected by SEM-BSE and EDS analyses in the Si-8at%Ti/GC system in a circular area surrounding the alloy drop and for a distance of around 20 μm. Conversely, moving far from the triple line, unreacted regions of GC appear with overlayered circular and narrowing areas of SiC. A less pronounced SiC layer was detected around the Si-24at%Ti alloy perimeter and even close to the Si-24at%Ti/GC triple line appearing as smaller crystals than those detected in the Si-8at%Ti/GC system (≈5 μm), as shown in the inserts of Figure 4a,b. Moreover, owing to the applied fast cooling of the two samples at the end of the wetting experiment, in full agreement with previous similar investigations, a crack at the interface was also observed. 

A different alloy microstructure was found at the top of the solidified drops (Figure 4c,d). Specifically, three well-distinguished developed microstructures/phases were detected at the top of the Si-24at%Ti alloy and consisting of theSi+TiSi_2_ eutectic phase, plus elongated crystals of pure Si embedded in a two-phase of pure Si + micro globular TiSi_2_–crystals colonies (Figure 4c). Contrarily, at the top of the solidified Si-8at%Ti/GC couple, except for a few TiSi_2_–micro crystals “rejected” at the grain boundaries of the eutectic phase, the latter microstructure was not revealed by SEM-BSE/EDS analyses (Figure 4d). On the other hand, elongated crystals of pure Si were found segregated at the alloy surface, as expected. 

The less pronounced growth of SiC crystals at the Si-24at%Ti/GC interface with respect to the Si-8at%Ti/GC is confirmed also by BSE-EDS analyses performed at the cross-sectioned sample after the wetting test, as shown in Figure 5. As can be seen, a compact layer of SiC with a thickness less than 2 μm is detected (Figure 5a). Moving toward the middle of the drop, a different microstructure was observed resulting in a thicker layer of SiC. In particular, it consists of a continuous plane (< 2μm), similar to the layer observed at the triple line, with stacked SiC crystals epitaxially grown up to 5–7 μm, as shown in Figure 5c. In the case of the Si-24at%Ti/GC solidified sample, a Si-TiSi_2_ two-phase microstructure consisting of globule-shaped TiSi_2_ crystals embedded in a pure Si matrix was observed in the alloy drop after solidification. In particular, an increase of TiSi_2_ crystal size was revealed moving from the middle of the drop (Figure 5c) to the triple line (Figure 5a). A comparable SiC layer was detected at the Si-8at%Ti/GC interface in terms of size and microstructure, as shown in Figure 5d. Although the SiC layer average thickness is comparable with that of the Si-24at%Ti/GC sample, at the Si-8at%Ti/GC triple line, a more compact layer is observed as the result of an enhanced SiC crystal packaging phenomenon (Figure 5b). Contrarily, the alloy microstructure at the metal-ceramic interface totally differs with respect to the Si-24at%Ti/GC couple. In fact, in the Si-8at%Ti/GC sample, a two-phase system is found after the solidification resulting in pure Si crystals surrounded by the Si+TiSi_2_ eutectic phase (Figure 5b,d).

### 3.2. Wettability of GC by Si-24at%Ti Alloys as a Function of the Testing Method

As previously mentioned, to check the reliability of the results obtained and to investigate the factors affecting the spreading kinetics, such as nonhomogeneous starting materials and the presence of pollutants, a targeted wetting experiment was performed at *T* = 1450 °C by dispensing a liquid Si-24at%Ti alloy through an Al_2_O_3_-capillary onto the GC plate. 

Aiming to achieve the complete alloy homogenization by melting TiSi_2_ precipitates (Figure 1d), the metal material inside the capillary was overheated up to 1500 °C for a few minutes, then the testing temperature of *T* = 1450 °C was imposed and the alloy squeezed onto the GC plate. Figure 6 shows the time sequence of the more relevant images recorded during the alloy squeezing from the capillary (1–4), the melt deposition on the GC substrate (5–7), and the wetting test (8–11). 

Although the image recording rate was 10 frames/s, just after the alloy detachment from the capillary, a contact angle value of θ ≈ 37° was shown at the newly formed Si-24at%Ti/GC interface, and the mentioned value was kept constant until the end of the experiment (*t* = 15 min), as shown in Figure 6. Similar to the sample processed by the classical sessile drop method, three well-distinguished microstructures were detected at the top of the Si-24at%Ti/GC solidified couple (Figure 7b) and consisted of a Si-TiSi_2_ two-phase microstructure embedded in a Si matrix, plus globule-shaped TiSi_2_ crystals segregated at the alloy surface. 

Beyond the triple line, residual amounts of alloy (Figure 7a,c), as well as a thin layer of nanometric SiC crystals, were detected on the GC substrate. A well-compacted layer of nanometric SiC crystals was also detected at the triple line (Figure 7d,e), in addition to overlayered colonies of SiC crystals epitaxially grown up to a size of 2–7 μm.

At the Si-24at%Ti/GC interfaces obtained by the two different methods, the same developed microstructure was mostly observed (Figure 8a,b). In both cases, globular-shaped TiSi_2_ crystals were observed in the alloy (close to the alloy/GC interface line) as “embedded” in pure Si and in a Si-rich Si-Ti eutectic phase moving forward to the top of the alloy drop. 

Similar to the sample obtained by the classical sessile drop method, a SiC layer was detected as the unique reaction product at the center of the interface (Figure 8d). In particular, a 2 μm SiC compact layer covered by SiC epitaxial crystals (size 2–7 μm) was again observed. The same microstructure was noticed at the triple line of the Si-24at%Ti/GC sample (Figure 8c). 

## 4. Discussion 

As introduced, the paper aims to study the feasibility of using Si-rich Si-Ti alloys for promoting the infiltration process in manufacturing SiC/TiSi_2_ composites. In addition to their low weight, such advanced materials are known for their high oxidation/corrosion resistances.

It is well known that a very good wetting (i.e., θ << 90°) between the infiltrating alloy and the porous material is one of the key requirements for making melt infiltration evolve spontaneously and, accordingly, for a successful process for the costless fabrication of nearly net-shaped composites. 

In this regard, as evinced by the new results presented in this work on the wettability of GC by liquid Si-rich Si-Ti alloys at *T* = 1450 °C under an Ar atmosphere, the alloys investigated are excellent candidates as infiltrating metal materials for C- and SiC-based porous systems to fabricate SiC/TiSi_2_ composites by the RFSC process. Indeed, a very good wetting was shown at the Si-Ti alloy/GC triple lines, as detailed in Table 1. In addition, for all the systems investigated, SiC was detected as the unique reaction product (Figure 5 and Figure 8), which is in agreement with the minimum content of Si (X^Si^_eq_) in the alloy enabling the spontaneous infiltration mechanism controlled by reactivity. Specifically, the X^Si^_eq_ is the composition for which all phases involved (Si–Ti/SiC/C) are in equilibrium. In fact, the presence of SiC at the interface is the primary condition for achieving a good wetting between liquid Si-based alloys and the C-based substrate [43]. Consequently, X^Si^_eq_ is one of the criteria for selecting the suitable range of Si-based alloy compositions to succeed in the fabrication of composites by the reactive infiltration process, and it can be predicted by using thermodynamic calculations [48]. On the other hand, as infiltration proceeds, the alloy composition may shift towards a composition richer in the alloying element, which might affect the infiltration kinetics due to the reduced reactivity or to the appearance of unexpected competitive reactions between the Si-alloying element and the C-based porous material.

An X^Si^_eq_ value of 0.69 was calculated for the Si-Ti/SiC/C system at *T* = 1450 °C, which is approaching the Si content of the TiSi_2_ alloy (X^Si^ = 0.66). Similar confirmation can be obtained by analyzing the recent studies reported by Roger et al. on Si-Ti-C [27,28] based on thermodynamic calculations using Thermo-Calc software [49] and the thermodynamic description reported by Du et al. [50]. As it can be seen by the ternary phase diagram calculated for the Si-Ti-C system at *T* = 1450 °C, biphase equilibria are predicted for a Si content less than 0.66 in the Si-Ti alloy compositions. Namely, for Si-Ti alloys with a Si content > 0.66, the coexistence of Si, TiSi_2_, and SiC is predicted, as confirmed by the results presented here.

After 15 min, the contact angle values shown at the Si-Ti alloy/GC triple lines at *T* = 1450 °C are comparable and in agreement with other similar Si-rich systems previously investigated [24,25,26,30,36,37,43,44,45,46,47]. On the other hand, a general trend of decreasing in the final contact angle measured after 15 min with the increasing of Si content is noticed. However, taking into account the experimental error, such a tendency is less pronounced in the composition range from eutectics to pure Si, and the final contact angle is almost coincident, as reported in Table 1. Contrarily, analyzing the wetting kinetics (U_wett_) results in U_wett_ (Si-24at%Ti) < U_wett_ (Si-16.2at%Ti) < U_wett_ (Si-8at%Ti). Such outcomes seem in contradiction with similar results reported in the literature [44], where a weak decrease in the spreading kinetics was observed with the increase of Si content in the Si-Co system processed under the same experimental conditions. As a general consideration, the increase of Si should increase the Si-based alloy reactivity towards GC, resulting in stronger C-dissolution and reactivity. Indeed, after 60 min, a thicker layer of SiC was observed at the Si-Co alloy richer in Si. On the other hand, the growth and thickening of SiC at the interface are time-dependent phenomena, which is more evident for a time of contact longer than 15 min, as documented in [45,51].

As known, the early stage of spreading observed during the classical sessile drop wetting experiments may be affected by the melting stage, which in turn can be influenced by (a) the presence of a primary oxide layer (SiO_2_) at the alloy surface, which is subsequently dissolved in Si monoxide by the reaction with Si [36] and by (b) a nonhomogeneous starting composition of the melting phase. Indeed, the presence of higher melting TiSi_2_ precipitates (*Tm* = 1485 °C [32]), embedded in the eutectic phase as detected for both the Si-16.2at%Ti [37] and Si-24at%Ti alloys (Figure 1), might be the reason of the delay observed in achieving the liquid state (Figure 3). Consequently, interaction phenomena may take place at the newly forming Si-Ti alloy/GC interfaces before the complete alloy melting.

In agreement with previous results obtained by processing the Si-16.2at%Ti/GC system under the same experimental conditions, strong Si evaporation/condensation phenomena were observed, mainly in the Si-8at%Ti/GC sample [36,37], as shown in Figure 4. Although processed at the same temperature, the less pronounced Si evaporation, resulting in the presence of a thin layer of SiC beyond the triple line, as observed in the Si-24at%Ti/GC couple, may be explained by difficulties encountered in achieving the steady-state conditions at the interface. In addition, the absence of individual epitaxially grown SiC crystals was noticed at the triple line of the cross-sectioned Si-24at%Ti/GC sample (Figure 5). Contrarily, the typical developed microstructure of the reaction product consisting of a first layer of SiC with more than well-distinguishable SiC crystals was found at the interface close to the middle of the drop. Such experimental observations led us to conclude that during the early stage of spreading, owing to the delay in achieving the complete melting and in crossing the transition limit between no-wetting to wetting behavior (i.e., θ = 90°), strong interfacial phenomena were already taking place. The absence of SiC crystals at the triple line may be due to the Si consumption occurring during the alloy spreading. As a consequence, the reduced Si content at the triple line with respect to the interface located at the central part of the Si-24at%Ti/GC sample might have affected diffusion phenomena between the alloy and the substrate. The poor Si content at the triple line may even be the reason for the higher contact angle value measured at the triple line at the end of the wetting test, as well as the precipitation of TiSi_2_ globular crystals during the solidification with a size bigger than those observed in the central part of the Si-24at%Ti/GC couple. Contrarily, after the wetting test performed by the dispensed drop method under the same experimental conditions, a Si+TiSi_2_ two-phase was observed in the Si-24at%Ti alloy in contact with GC both at the triple line and moving to the central part of the sample (Figure 8). However, as can be seen in Figure 6 and Figure 7, the presence of a residual amount of alloy material on the GC close to the triple line provides evidence that a dewetting phenomenon most probably occurred after the alloy deposition onto the GC substrate. This is caused by the alloy perturbation when it is placed onto a substrate and its subsequent detachment by moving up the capillary [37]. On the other hand, dewetting is mainly observed at the interface of negligible or less reactive systems. The lower reactivity of the liquid Si-24at%Ti alloy with respect to GC could be explained by a non continuous thin layer of SiC already present on GC and formed during the approaching of the pendant drop to the substrate. SiC is well wettable by Si-based alloys with respect to C and responsible for the decreased contact angle values measured at the Si-24at%Ti/GC triple line (Table 1), as observed in [46]. As confirmation, a well-adherent GC/SiC/Si-based alloy system is observed at the end of the wetting tests. The strong adherence combined with the mismatch in the GC/SiC/Si-silicide CTE values is also responsible for the crack noticed at the interface.

## 5. Conclusions

To succeed in the fabrication of SiC/TiSi_2_ lightweight composites by reactive infiltration, the feasibility of using Si-rich Si-Ti alloys as infiltrating alloys was studied for the first time. In particular, the Si-Ti alloys investigated seem to be excellent candidates for infiltrating metal materials for C- and SiC-based porous systems to fabricate SiC/TiSi_2_ composites by the RFSC process.

Specifically, based on the very good wetting (i.e., θ << 90°) observed at the Si-Ti alloy/GC triple lines at *T* = 1450 °C, the Si-rich Si-Ti alloys selected meet the key requirements in promoting reactive infiltration evolving by a spontaneous mechanism. In addition, the Si-content seems to weakly affect the spreading kinetics mainly due to nonhomogeneous starting alloy materials. Such a conclusion is substantiated by the results obtained from squeezing the alloy through the capillary. The absence of primary oxide at the alloy surface and an improved homogenization were crucial for obtaining a regular Si-24at%Ti/GC interface.

In full agreement with thermodynamics and similar studies reported in literature, SiC as a unique reaction product, was detected at the interface. 

In this regard, the compactness of the reaction layer is enhanced by the increase of Si content in the alloy composition and the Si condensation/evaporation phenomena observed beyond the advancing triple line.

As a general consideration, aiming to improve the overall thermomechanical response of the produced composite by replacing/reducing Si in the metal phase, a Si content larger than the eutectic composition is not recommended.

## Figures and Tables

**Figure 1 materials-14-03746-f001:**
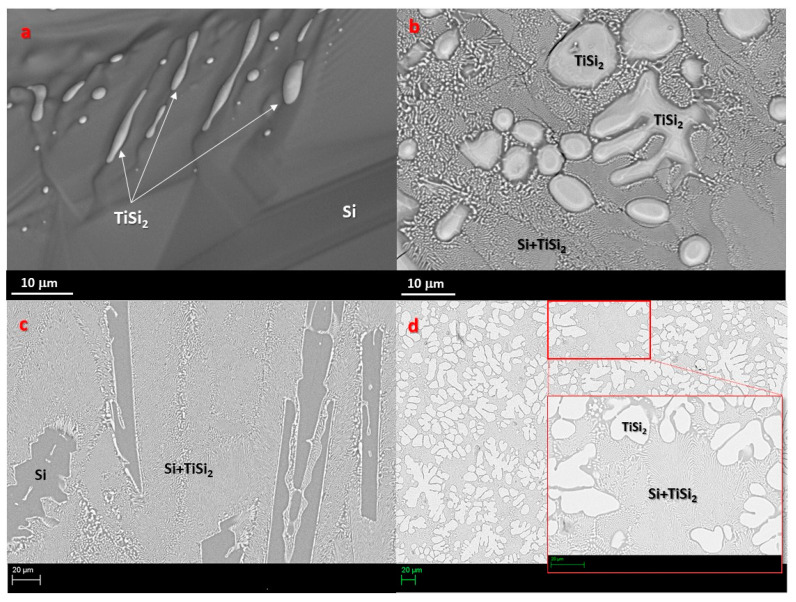
SEM-BSE images and phase identification by EDS analyses performed both at the (**a**,**b**) top of the solidified and at the (**c**,**d**) cross-sectioned Si-8at% Ti and Si-24at% Ti alloy samples after arc melting.

**Figure 2 materials-14-03746-f002:**
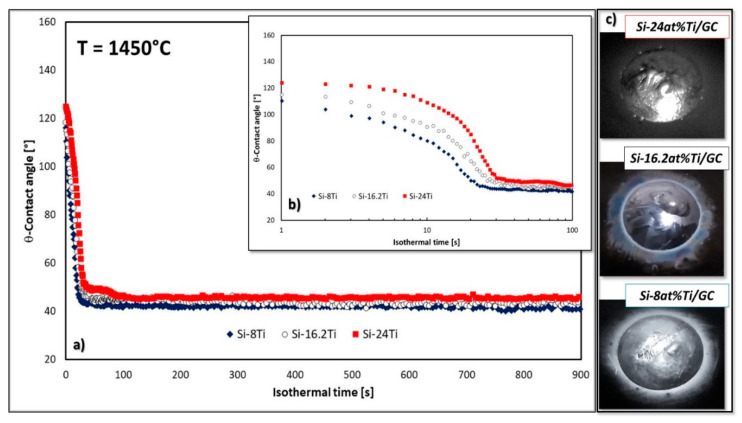
Contact angle behaviors at *T* = 1450 °C under an Ar atmosphere as a function of Si-content observed after (**a**) 900 s and (**b**) 100 s; (
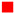
) Si-24at%Ti /GC, (◯) Si-16.28at%Ti/GC, (
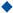
) Si-8at%Ti/GC; (**c**) related top view photographs of Si-Ti alloys/GC couples after the wetting experiments.

**Figure 3 materials-14-03746-f003:**
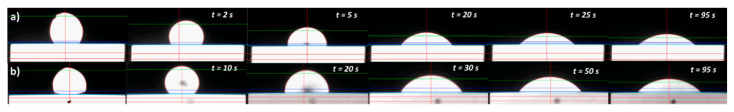
Time sequence images recorded during the wetting experiments performed by sessile drop method at *T* = 1450 °C under an Ar atmosphere; (**a**) Si-8at%Ti/GC and (**b**) Si-24at%Ti/GC.

**Figure 4 materials-14-03746-f004:**
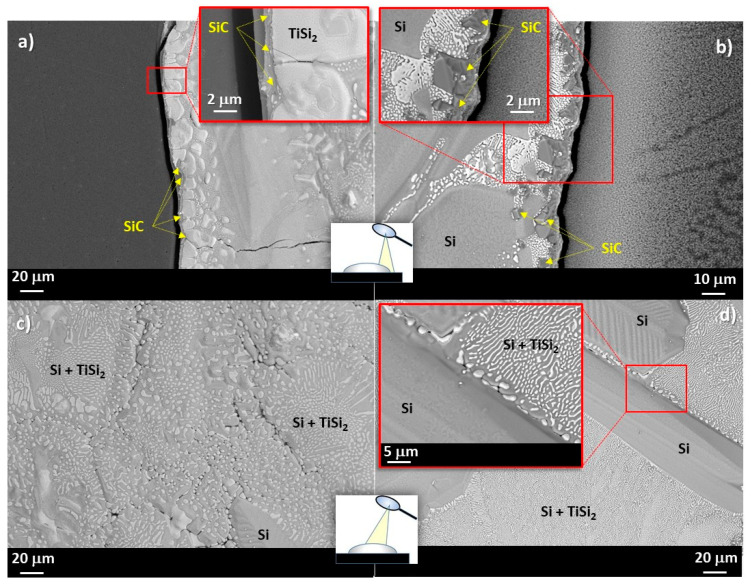
SEM-BSE images and proposed phase identification by EDS analyses performed at different magnifications at the triple lines and at top of the (**a**,**c**) Si-24at%Ti/GC and (**b**,**d**) Si-8at%Ti/GC samples after the wetting tests performed at *T* = 1450 °C for 15 min.

**Figure 5 materials-14-03746-f005:**
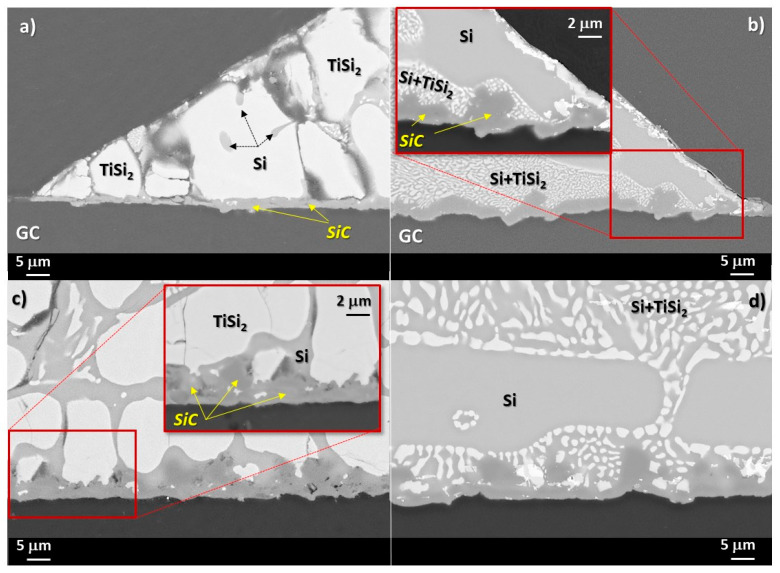
SEM-BSE images and proposed phase identification by EDS analyses performed at different magnifications at the triple lines and at the interfaces of the (**a**,**c**) cross-sectioned Si-24at%Ti/GC and (**b**,**d**) Si-8at%Ti/GC samples after the wetting tests performed at *T* = 1450 °C for 15 min.

**Figure 6 materials-14-03746-f006:**
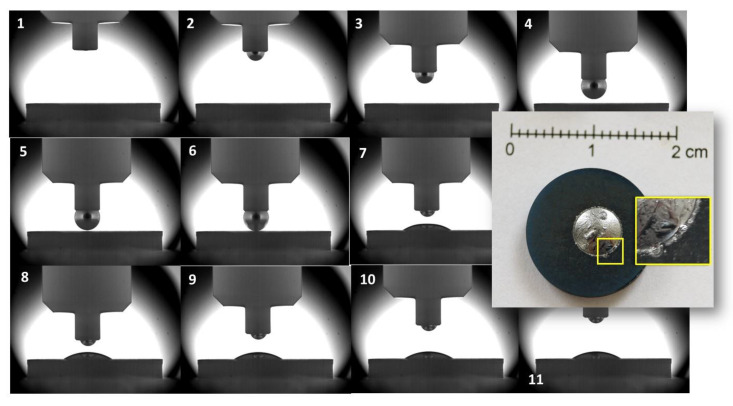
Post-mortem pictures of the Si-24at%Ti/GC sample and the time sequence of images recorded during the wetting test performed by the dispensed drop method at *T* = 1450 °C under an Ar atmosphere.

**Figure 7 materials-14-03746-f007:**
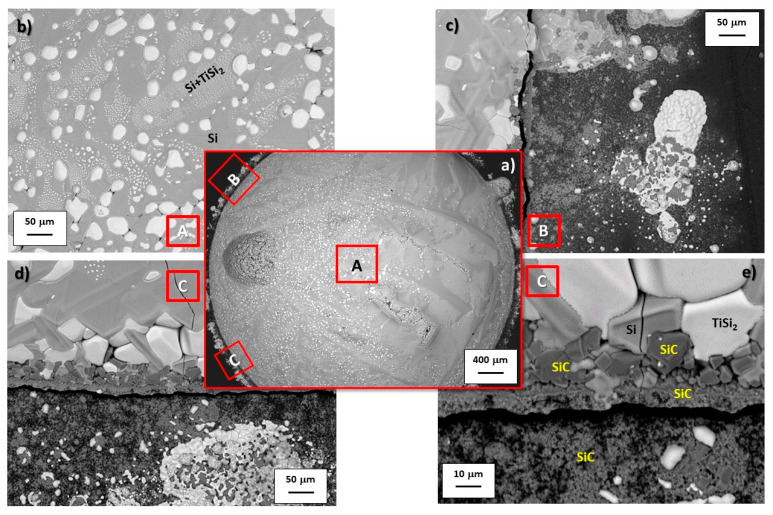
SEM-BSE and proposed phase identification by EDS analyses performed at different magnifications at the top (**a**,**b**) and at the triple line (**c**–**e**) of the solidified Si-24at%Ti/GC sample after the wetting test performed by the dispensed drop method at *T* = 1450  °C under an Ar atmosphere. A: top drop; B and C Different areas at the triple line.

**Figure 8 materials-14-03746-f008:**
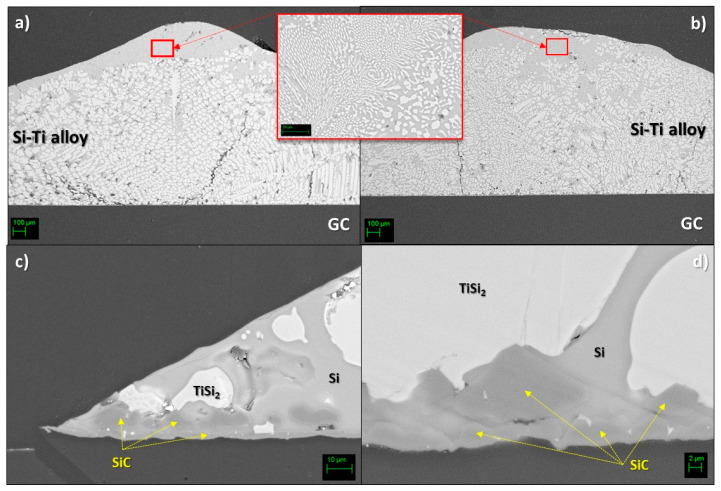
SEM-BSE images and phase identification by EDS analyses performed at different magnifications at the cross-sectioned Si-24at%Ti/GC samples processed by (**a**) classical sessile and (**b**–**d**) dispensed drop methods at *T* = 1450 °C under an Ar atmosphere.

**Table 1 materials-14-03746-t001:** Equilibrium contact angle values (θ_f_) measured for the liquid Si-rich Si-Ti alloys in contact with GC and SiC at *T* = 1450 °C under an Ar atmosphere by the sessile drop (SD) and dispensed drop (DD) methods.

System	Method	*T* (°C)	*t* (min)	θ_f_ ± 2 (°)	Reference
Si/GC	SD			38	[47]
Si-8at%Ti/GC	SD	1450	15	42	This work
Si-16at%Ti/GC	SD			43	[36,37]
Si-24at%Ti/GC	SD			46	This work
Si-16at%Ti/GC	DD			41	[36]
Si-24at%Ti/GC	DD			37	This work
Si-16at%Ti/SiC	SD			44	[37]

## Data Availability

The data presented in this study are available on request from the corresponding author.

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
