# Peer review of "Interface Design in Lightweight SiC/TiSi2 Composites Fabricated by Reactive Infiltration Process: Interaction Phenomena between Liquid Si-Rich Si-Ti Alloys and Glassy Carbon"

_materials, 2021, doi:10.3390/ma14133746_

Round 1

Reviewer 1 Report

In forming composite materials, the properties of the interface between the adjoining materials is exceedingly important.  In this manuscript, the authors conduct challenging high-temperature measurements of Si-rich Si-Ti alloy wetting behaviour of glassy carbon, and analyse the reactive spreading and the microstructure of the interface that is formed.  The experiments, data analysis and discussion are of high quality, and will contribute significantly to the the better understanding and further development of this important class of materials.

To increase the scholarly merit of the work further, I ask the authors to consider the possible effect of friction forces during the reactive spreading of the liquid drop (see Nature 534, 2016, p. 676) on the kinetics and final contact angles that are measured, and for which the experimental circumstances are quite different from aqueous liquid–solid interfaces.

Author Response

In forming composite materials, the properties of the interface between the adjoining materials is exceedingly important.  In this manuscript, the authors conduct challenging high-temperature measurements of Si-rich Si-Ti alloy wetting behaviour of glassy carbon, and analyse the reactive spreading and the microstructure of the interface that is formed.  The experiments, data analysis and discussion are of high quality, and will contribute significantly to the the better understanding and further development of this important class of materials.

The authors wish to thank the Reviewer for her/his positive comments on the quality of work submitted, mainly considering how much are challenging high-temperature experimental studies on such highly unstable alloys at the liquid state.

To increase the scholarly merit of the work further, I ask the authors to consider the possible effect of friction forces during the reactive spreading of the liquid drop (see Nature 534, 2016, p. 676) on the kinetics and final contact angles that are measured, and for which the experimental circumstances are quite different from aqueous liquid–solid interfaces.

The authors appreciated very much the Reviewer’s suggestion to drive the discussion and even the intepretation of our results in view of possible effects of friction forces during reactive spreading. Indeed, other experiments, mainly supported by an improved optical line consisting in a different lighting source combined with more efficient IR filters, as well as a higher resolution-speed camera, are planned. In addition, to identify properly the transition between advancing-receiding angles an increased fragmentation (n. of pixels) of the images is crucial. The digitalization applied for the experiments here presented was 512x512, then too low for detecting advancing/receiding phenomena at the triple lines. It doesn’t mean that we consider as negligible such phenomena. We believe that friction and sticking phenomena are taking place as driven/resulting from the growing of the SiC layer at the interface, mainly as epitaxial crystals. For these reasons, as aforementioned and insipired by the paper suggested by the Reviewer, the authors will perform experiments with such motivation behind.

Reviewer 2 Report

This manuscript investigates the wetting behavior of Si-reich Si-Ti alloys on glass carbon surface to evaluate the feasibility of using such alloy for composite fabrication. The idea is very interesting but the experimental part is relatively thin. I recommend for publication with more content incorporated. Some other comments,

  1. Why choose 1450C? I understand that working temperature needs to be less than 1450C but what about a slightly lower temperature such as 1420C?
  2. Page 6, Figure 3. To prove that there is not much change of contact angle before and after quenching, an image of before / after quenching could be displayed.
  3. Only Si-8at%Ti and Si-24at%Ti SEM images displayed here. What about the 16.2 composition?
  4. Page 6, line 264 and Page 7 line 268, the micro unit sign was lost after conversion in the pdf manuscript.

Author Response

This manuscript investigates the wetting behavior of Si-reich Si-Ti alloys on glass carbon surface to evaluate the feasibility of using such alloy for composite fabrication. The idea is very interesting but the experimental part is relatively thin. I recommend for publication with more content incorporated. Some other comments,

  1. Why choose 1450C? I understand that working temperature needs to be less than 1450C but what about a slightly lower temperature such as 1420C?

The authors fully agree with the Reviewer’s comment. In fact, mainly keeping in mind the real scope of the paper consisting in providing evidences about the effect of process parameters in “degrading” the GC substrate, which in turn can be related to the fiber degradation, the decrease of temperature should benefit in decreasing the reactivity and the overall interaction phenomena occurring between the liquid Si-Ti alloys and GC. On the other hand, aiming to “mimic” the working conditions imposed during industrial processes, we selected T = 1450°C which is the typical temperature imposed during infiltration processes. Actually, even if the temperature can be decreased down to T = 1420°C (6 degrees higher than the melting temperature of pure Si), exothermic phenomena may be occurred at the interface, with a local increase of temperature, due to the reaction of liquid Si with C to form SiC. 

  1. Page 6, Figure 3. To prove that there is not much change of contact angle before and after quenching, an image of before / after quenching could be displayed.

The authors wish to thank the Reviewer for her/his valuable suggestion. It will be surely done for future experiments. For the present results, we can not provide images after solidification since no any image was recorded after the end of dwell time 

  1. Only Si-8at%Ti and Si-24at%Ti SEM images displayed here. What about the 16.2 composition?

Since the results on Si-16.2at%Ti have been already published, we decided to drive the focus on the new alloy compositions and to cite the papers where the results on the eutectic alloy are reported

  1. Page 6, line 264 and Page 7 line 268, the micro unit sign was lost after conversion in the pdf manuscript.

The authors wish to thank the Reviewer for her/his suggestion/remark and we apologize for that. A careful check of the manuscript has been made and all the typos removed.

Hopefully it will be appreciated, the english grammar has been checked and corrections/changes are made in the manuscript (in red).

Reviewer 3 Report

Thanks for your submission to Polymers. The manuscript should be revised carefully based on below comments:

  1. I am not very convinced regarding the novelty of the work. Please add in the novelty quotient of your work at the end of the introduction section.
  2. Line 162, A strong Si-segregation was detected at the top of the Si-8at%Ti solidified, not very clear, please rephrase.
  3. Line 236, any justification? Any similar evidence from the literature.
  4. Line 272, not very clear, rephrase.
  5. Line 311, avoid using the words like affecting factors, better is factors affecting the...
  6. Check line 356
  7. Overall the results are very clear and the manuscript is easy to read. But it requires significant English and grammar check.
  8. Please precise the conclusion and write down the salient findings. 

Author Response

The manuscript should be revised carefully based on below comments:

  1. I am not very convinced regarding the novelty of the work. Please add in the novelty quotient of your work at the end of the introduction section.

Following the Reviewer’s suggestion, although it was already explained, the novelty of the work has been repeated and clarified much more in the Introduction section.

  1. Line 162, A strong Si-segregation was detected at the top of the Si-8at%Ti solidified, not very clear, please rephrase.

As suggested, the sentence has been rephrased as:

A strong Si-segregation was detected at the top of the Si-8at%Ti solidified drop (Figure 1a). At the Si-8at%Ti cross-sectioned sample, pure Si in the form of elongated needle-shaped crystals embedded into a Si-Ti eutectic phase, was revealed (Figure 1c).

  1. Line 236, any justification? Any similar evidence from the literature.

In answering to the Reviewer’s comment, in line 236, the results are described. The phenomena observed are deeply discussed in the section n. 4 entitled “Discussion” where similar evindences from literature are cited.

  1. Line 272, not very clear, rephrase.

As suggested, the entire sentence has been rephrased as following:

“Moving toward the middle of the drop, a different microstructure was observed and resulting in a thicker layer of SiC. In particular, it consists of a continuous plane (< 2 mm), similar to the layer observed at the triple line, with stacked over SiC crystals epitaxially grown up to 5-7 mm, as shown in Figure 5c.”

  1. Line 311, avoid using the words like affecting factors, better is factors affecting the...

Following the Reviewer’s suggestion, sentences like “affecting factors” have been changed as “factors affecting the….”

  1. Check line 356

To avoid confusion, the equation has been removed.

  1. Overall the results are very clear and the manuscript is easy to read. But it requires significant English and grammar check.

We are thankful for the comment about the clear description of the results.

In fully agreement with the Reviewer’s comment/suggestion, the manuscript has been carefully checked and english grammar revised (in red).

  1. Please precise the conclusion and write down the salient findings. 

As suggested, the conclusion section has been rewritten focusing and put in evidence only the salient findings.
